# Repetitive concussions promote microglia-mediated engulfment of presynaptic excitatory input associated with cognitive dysfunction

Maryam Chahin[1,2,3], Julius Mutschler[1,2,5], Stephanie P. Dzhuleva[1,2,5], Clara Dieterle [1,2], Leidy Reyes Jimenez[1,2], Srijan Raj Bhattarai[1,2], Valerie Van Steenbergen[1,2] & Florence M. Bareyre [1,2,4] ✉

Concussions are a current health concern and account for the vast majority of head trauma. While symptoms after a single impact are usually transient, repetitive concussions, as often occur in sports, are responsible for persistent acute and chronic deficits. Here, we used a model of bilateral midline-centered concussions in mice to show that repetitive concussions selectively induce impairments in learning ability compared to single-impact injuries. Since microglial cells and their activation are considered key factors in degenerative pathology after brain trauma, we examined their structure and function after single and repetitive concussions in the cortex underlying the concussions and in the hippocampus. We found that only repetitive concussions led to a significant long-lasting structural activation of microglia and an increase in microglia-mediated engulfment of presynaptic excitatory synapses, while the elimination of inhibitory synapses was not altered. Since the density of excitatory input did not change during the 6-week study period, we hypothesize that there is a turnover of excitatory synapses following repetitive concussion that can be compensated for, anatomically but not behaviorally.

Traumatic brain injury (TBI) is one of the most debilitating causes of disability in young adults and children, affecting approximately 70 million people worldwide each year[1]. Among all forms of TBI, concussions, often referred to as "mild TBI", account for approximately 80% of these head injuries[2,3]. Although these concussions usually result in short-term behavioral deficits, in some cases these deficits can be observed over several months[4,5]. An important factor in the development of long-term deficits is the repetition of these concussions, especially if they occur before the brain has fully recovered from the previous injury[6], which is particularly common in contact sports[7]. While exercise has long been recognized for its numerous physical and mental health benefits, it is increasingly recognized that repetitive concussions can pose a significant risk to brain health[8]. In particular, several clinical reports have shown that individuals involved in these repetitive sports-related concussions may experience short-, medium- and long-term consequences of these repetitive head impacts, ranging from immediate and short-term loss of cognitive function and mood changes to

long-term physiological changes such as dementia, progressive tauopathy or chronic encephalopathy[9,10]. Treatment of these sports-related repetitive concussions is complicated, as current return-to-play guidelines classically assume that symptom-free behavior equates to a return to brain homeostasis and loss of cellular dysfunction. However, many clinical studies have shown that most people with repetitive concussions continue to have symptoms for at least a week after the blows[11], and recent reports even suggest that symptoms worsen over time[12,13]. Currently, there is no treatment for single and repetitive concussions, partly because people with concussions never visit a hospital, and partly because patients who do go to the hospital are left to heal their injuries and only the symptoms are treated, not the causes[9].

In the central nervous system (CNS), microglia, the resident immune cells, are responsible for constant monitoring of the microenvironment to clear debris, maintain homeostasis, and protect neuronal function[14,15]. In case of CNS damage, microglia can become activated and have a dual role in

---

[1]Institute of Clinical Neuroimmunology, University Hospital, LMU Munich, Munich, Germany. [2]Biomedical Center Munich (BMC), Faculty of Medicine, LMU Munich, Planegg-Martinsried, Germany. [3]Graduate School of Systemic Neurosciences, LMU Munich, Planegg-Martinsried, Germany. [4]Munich Cluster of Systems Neurology (SyNergy), Munich, Germany. [5]These authors contributed equally: Julius Mutschler, Stephanie P. Dzhuleva. ✉e-mail: florence.bareyre@med.uni-muenchen.de

regulating inflammation[16]. On the one hand, microglia are pivotal in removing cellular debris, in the secretion of neurotrophic factors and anti-inflammatory cytokines, and in tissue healing[17]. On the other hand, microglia, when dysregulated following CNS injury, become key players in the development of chronic neurodegenerative pathology and can then contribute to further exacerbation of damage[17]. Although they have been observed to proliferate after repetitive concussions, the role of microglial cells in this context is not yet fully understood, and it is not clear what contribution repetitive concussions make to their activation. Therefore, in this study, we performed single and repeated concussions in mice and were able to show that our model mimics the clinical situation without motor deficits but with significant cognitive and learning impairments over time. We then investigated the structural and functional adaptations of microglia from acute to chronic time points after injury. We show specific structural and functional activation of microglia after repeated concussions, associated with selective microglia-mediated engulfment of presynaptic excitatory inputs in the cortex and hippocampus. This engulfment was accompanied by an initial loss of excitatory synapses that later recovered, suggesting that repeated concussions trigger a specific engulfment of excitatory synapses. In contrast, inhibitory input was not affected by the single or repeated concussions. In conclusion, we highlight the contribution of microglial cells to the specific engulfment of excitatory input following repetitive concussions in mice and suggest that this initial loss of synapses can be compensated for, at anatomical levels but not behaviorally.

## Results

### Repetitive concussions specifically lead to learning disorders without changing motor behavior and without cell death

To determine the anatomical and functional consequences of repetitive concussions, we used a controlled cortical impact device with a custom-made, flexible silicone tip with a diameter of 5 mm and a hardness of 55 Shore A. We impacted the sensorimotor cortex, bilaterally across the midline spanning the cortex with lambda as the caudal landmark, with either a single concussion or three concussions 48 h apart[18] (Fig. 1a, b). We could not detect any macroscopic changes in the brains in the sense of hematomas, hemorrhages, edema, or bruising (Fig. 1c). To evaluate the effects of repetitive concussion on acute physiological responses and to determine whether apnea and righting reflex depend on the repetition of concussion, the responses of mice exposed to up to three repetitive concussions 48 h apart were compared. The duration of apnea decreased as a function of the number of concussions (Supplementary Fig. 1a; p = 0.0003 one-way ANOVA and Dunnett's test 1st vs 3rd concussion) and the righting reflex, i.e. the time required for the mice to spontaneously roll back to a prone position was significantly reduced with the number of blows and especially after the third concussion (Supplementary Fig. 1b; p < 0.0001 ANOVA and Dunn's test 1st vs 3rd contusion and p = 0.0479 2nd vs 3rd contusion). These physiological responses are consistent with previous data on repetitive concussion[18]. We monitored the weight of the mice after the repetitive injuries and found no statistical differences (Supplementary Fig. 1c). Although we could not detect any macroscopic changes at the brain surface after the single and repetitive concussions, we investigated whether cell death occurred following the impact(s). To this end, we performed two analyses. First, we counted the number of neurotrace-positive cells in layer II/III of the cortex (directly under the concussion) at 1 week, 3 weeks, and 6 weeks following the concussion(s), and second, we stained for activated caspase-3 as a marker of apoptotic cell death at 2 d, 1 week, 3 weeks, and 6 weeks following the concussions. We did not detect a decrease in the number of neurotrace-positive cells or an increase in the number of activated caspase-3-positive cells compared to control mice, suggesting that our model is very mild and does not induce cell death in layer II/III of the cortex below the impact at the time points examined (Supplementary Fig. 1d, e). We also investigated cell death in the hippocampus at 2 days post-concussion and could similarly not detect any staining.

Finally, we wondered whether the consequences of these single and repetitive concussions could be distinguished at the behavioral level. To this

end, we used four behavioral tests to assess motor and cognitive function after single and repetitive concussions. To evaluate the motor function, we performed the rotarod to test balance and endurance and the ladder rung test to assess locomotion[19]. We found no differences in any of the parameters tested between sham mice, which performed as expected, and mice that suffered a single or repetitive concussion(s) (Supplementary Fig. 2). This emphasizes that gross motor function is not significantly altered after either a single concussion or repetitive concussions at acute and chronic time points.

To assess cognitive function, we used the Y-maze, which allows the evaluation of short-term memory and spatial working memory in mice (Fig. 1d). In this test, the mice are allowed to explore all three arms of the maze after one arm was previously blocked. The test is driven by the rodents' curiosity to explore previously unvisited areas[20]. We evaluated the preference index for the new arm and found that it was stable at 0.35 in control animals throughout the study period, which is consistent with the literature[21] (Fig. 1e). Interestingly, in mice exposed to a single concussion, we did not observe any changes over the study period. However, mice subjected to repetitive concussions showed a significant decrease in the preference index 1 week after injury (Fig. 1e; p = 0.0480, Two-way ANOVA followed by Tuckey's multiple comparison test sham 1 wk vs 3 concussions 1 wk). The deficits in the preference index, indicating that spatial memory was impaired in the mice that had received 3 concussions, were transient and recovered at week 3 post-injury. We also analyzed the time spent in the conditioning arm. Mice that had sustained a single concussion showed no differences from the sham mice and were stable throughout the study period (Fig. 1f). In contrast, mice that had suffered repetitive concussions spent significantly more time in the conditioning arm after 1 and 6 weeks, indicating memory impairment (Fig. 1f; p = 0.0230 sham 1 wk vs 3 concussions 1 wk and p = 0.0363 sham 6 wks vs 3 concussions 6 wks; two-way ANOVA followed by Tuckey's multiple comparison test). We then used the tail suspension test (Fig. 1g) to assess learned helplessness[22,23]. We recorded the percentage of total experimental time that mice spent struggling and the mean duration of the struggles to escape suspension. In the sham animals, we observed that the evaluated parameters steadily decreased over the experimental period, indicating that the animals learn that there is no escape (Fig. 1h, i; 2-way ANOVA followed by Tuckey's multiple comparisons tests. % of total time struggling: p = 0.0074 sham baseline vs sham 3 wks; p = 0.0016 sham baseline vs sham 6 wks; Total duration of struggles: p = 0.0073 sham baseline vs sham 3 wks; p = 0.0015 sham baseline vs sham 6 wks). Mice that had sustained a single concussion showed the same pattern as the sham mice, albeit slightly delayed and with a less pronounced decline, suggesting transient learning impairments (Fig. 1h, i; 2-way ANOVA followed by Tuckey's multiple comparisons tests. % of total time struggling: p = 0.0054 1 concussion baseline vs 1 concussion 6 wks; Total duration of struggles: p = 0.0053 1 concussion baseline vs 1 concussion 6 wks). However, mice that had suffered 3 consecutive concussions showed no such decrease and continued to struggle similarly to baseline (Fig. 1h, i). As this test is an indicator of learned helplessness, this may suggest that mice that have suffered three concussions show a marked impairment in learning that can be detected in the suspension test.

### Structural microglial changes show activation of microglia after repetitive concussions in the cortex underlying the impacts

Following repetitive concussions, microglia have been reported to proliferate and undergo microgliosis[24], and activated microglia are believed to affect normal homeostasis and be key mediators in the evolution of chronic trauma-induced neurodegeneration[25]. We focused all of our initial analysis on the sensory cortex underlying the impact (S1, layer II/III, −1.5 mm from Bregma) as we hypothesized that this should be the region most affected by the concussion and its recurrences. We first examined the phagocyte response and found a significant proliferation of Iba1+ cells – probably resident microglia and infiltrating macrophages – at 1, 3, and 6 weeks in the single concussion and at 1 and 3 weeks in the repetitive concussions (Fig. 2a, b). We then aimed to determine the structural changes in microglia

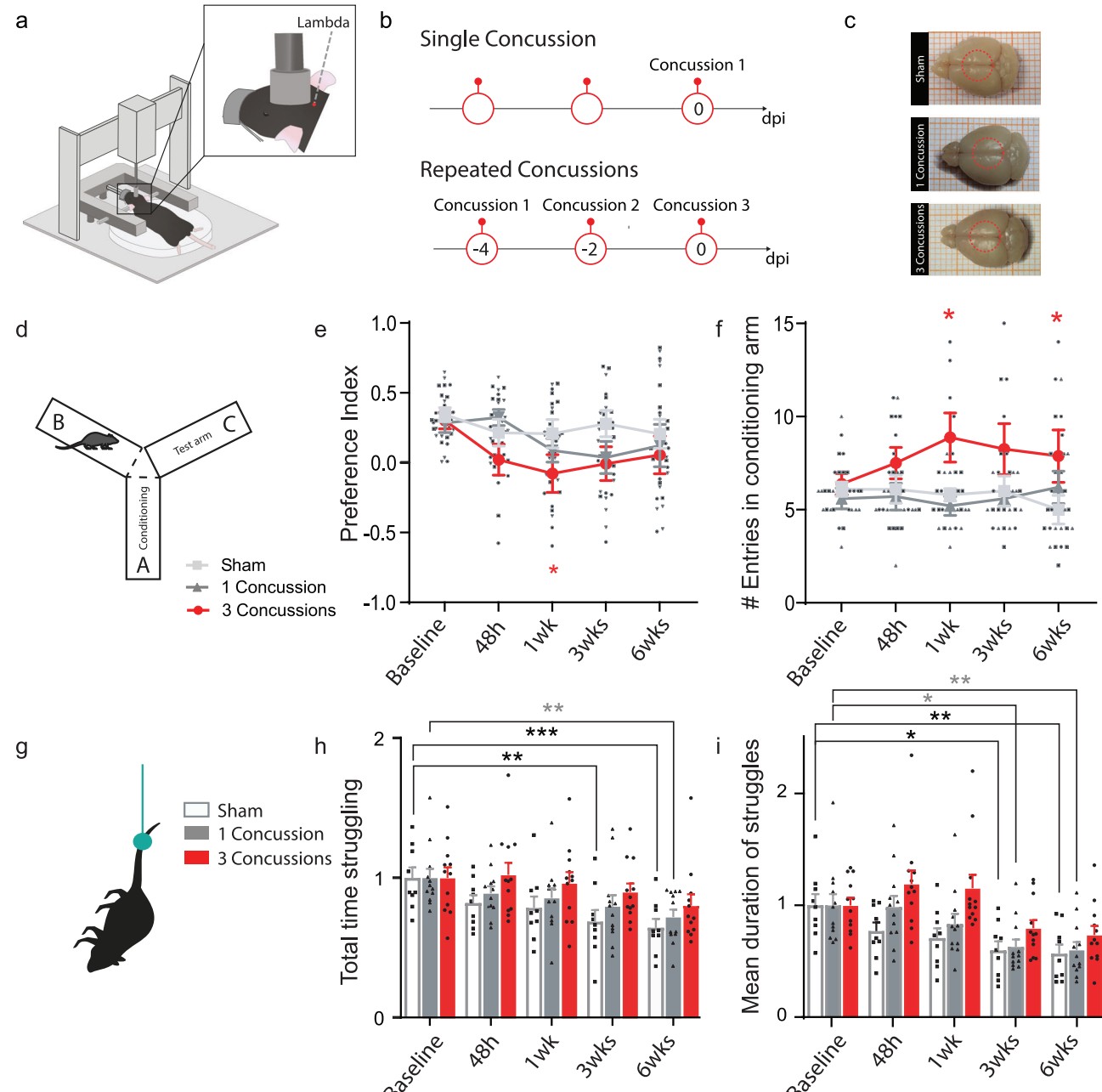

**Fig. 1 | Repetitive concussion triggers perturbation in learning abilities in mice.**
**a** Scheme of the controlled cortical impact device used to produce repetitive closed-skull concussions in mice. Note the location of the impact with the caudal landmark positioned on lambda. **b** Timeline of the experiment with the concussions delivered either single or repetitive at 48 h intervals. **c** Macroscopic view of the brains following sham injury, single concussion, or repetitive concussions. Red circles depict the impact size and location on the brain. **d** Schematic representation of the Y-maze test. **e** Quantification of the preference index in sham (light gray line), single concussion (dark gray line), and repetitive concussions (red line) at different time points following injury (48 h, 1 wk, 3 wks, 6 wks). p = 0.0480 (Two-way ANOVA followed by Tuckey's multiple comparison test sham 1 wk vs 3 concussions 1 wk). n = 8–10 animals per group. **f** Quantification of time spent in the conditioning arm in sham (light gray line), single concussion (dark gray line), and repetitive concussions (red line) at different time points following injury (48 h, 1 wk, 3 wks, 6 wks). p = 0.0230 sham 1 wk vs 3 concussions 1 w and p = 0.0363 sham 6 wks vs 3

concussions 6 wks (Two-way ANOVA followed by Tuckey's multiple comparison test). n = 8–10 animals per group. **g** Schematic representation of the tail suspension test. **h** Quantification of the percentage of total experimental time spent struggling in sham (white box), single concussion (gray box), and repetitive concussions (red box) at different time points following injury (48 h, 1 wk, 3 wks, 6 wks). p = 0.0074 sham baseline vs sham 3 wks; p = 0.0016 sham baseline vs sham 6 wks; p = 0.0054 1 concussion baseline vs 1 concussion 6 wks (Two-way ANOVA followed by Tukey's multiple comparison test). n = 9–12 animals per group. **i** Quantification of the mean duration of struggles in sham (white box), single concussion (gray box), and repetitive concussions (red box) at different time points following injury (48 h, 1 wk, 3 wks, 6 wks). p = 0.0073 sham baseline vs sham 3 wks; p = 0.0015 sham baseline vs sham 6 wks; p = 0.0053 1 concussion baseline vs 1 concussion 6 wks (Two-way ANOVA followed by Tukey's multiple comparison test). n = 9–12 animals per group. wk: week; h: hour. Data as mean ± SEM.

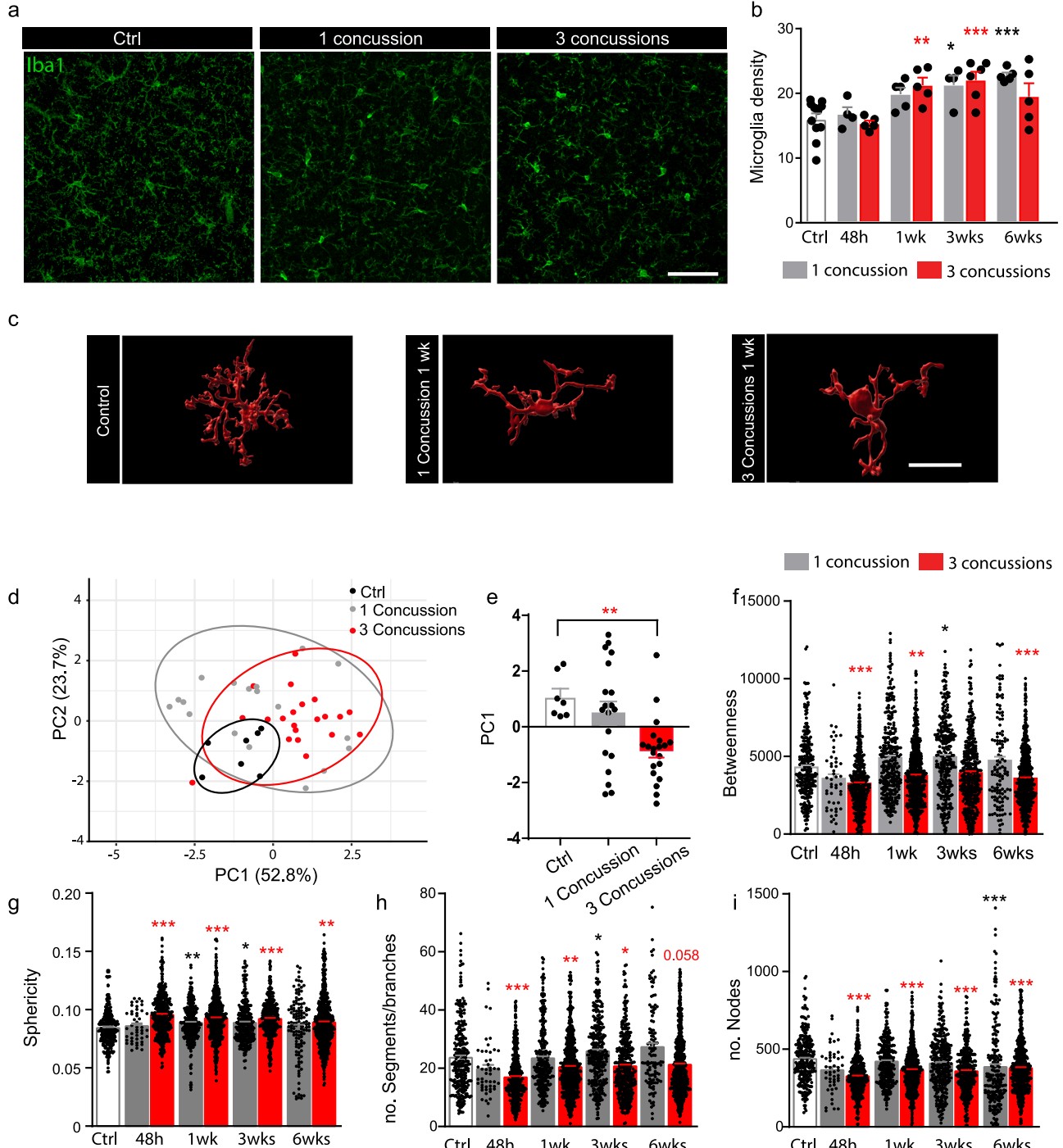

**Fig. 2 | Microglia become ameboid in the cortex following single and repetitive concussions. a** Representative confocal images of the cortex from control, 1 concussion, and 3 concussions mice immunostained against Iba1 (Iba1: green; blue: dapi) 1 week after the last injury. Scale bars: 100 μm. **b** Quantification of the number of Iba1+ cells in the cortex at different time points 48 h 1, 3, and 6 weeks following single (gray bars) or repetitive (red bars) concussions. n = 4–11 mice per group. p = 0.0082 3 concussions 1 wk vs Ctrl; p = 0.0165 1 concussion 3 wks vs Ctrl; p = 0.0008 3 concussions 3 wks vs Ctrl; p = 0.0004 1 concussion 6 wks vs Ctrl. **c** 3D Imaris representations of the morphological changes in microglia following concussion(s). Scale bars: 20 μm. **d** Principal component analysis of the morphological features to reduce dimensionality following single repetitive concussions. (**e**) Quantification of average scores on principal components 1 for control, 1 concussion, and 3 concussions. p = 0,0084 Control vs 3 concussions; one-way ANOVA followed by Dunnett's test. **f–i** Quantification of the microglia betweenness (**f**; p > 0.0001 Ctrl vs 3 concussions 48 h; p = 0.0092 Ctrl vs 3 concussions 1 wk; p = 0.023 Ctrl vs 1 concussion 3 wk; p > 0.0001 Ctrl vs 3 concussions 6 wks),

sphericity. (**g**; p < 0.0001 Ctrl vs 3 concussions 48 h; p = 0.0016 Ctrl vs 1 concussion 1 wk; p < 0.0001 Ctrl vs 3 concussions 3 wks; p00.0496 Ctrl vs 1 concussion 3 wks; p < 0.0001 Ctrl vs 3 concussions 3 wks;p = 0.0025 Ctrl vs 3 concussions 6 wks), number of segments per branches (**h**; p = 0.0001 Ctrl vs 3 concussions 48 h; p = 0.0082 Ctrl vs 3 concussions 1 wk; p = 0.0409 Ctrl vs 1 concussion 3 wks; p = 0.0488 Ctrl vs 3 concussions 3 wks) and number of nodes (**i**; p < 0.0001 3 concussions vs Ctrl for all time points) of microglia in the cortex at different time points 48 h 1, 3, and 6 weeks following single (gray bars) or repetitive (red bars) concussions. n = 4–7 mice per group and N = 45–692 microglia per group. Data represent mean ± SEM. Significant differences with *p < 0.05, ***p < 0.001. Scale bar equals 5 μm. All data were tested with the Shapiro–Wilk test. Data in (**b**) and (**e**) are normally distributed. Data in (**f–i**) are not normally distributed. Comparisons were then made using a one-way ANOVA and Dunnett's test in (**b**) and (**e**). Kruskall–Wallis test followed by Dunn's test was used for (**f–i**). Ctrl: Control; wk: week. Data as mean ± SEM.

following single and repetitive concussions to determine whether single and repetitive concussions can trigger their activation. To do so, we used a recently published automated morphological analysis toolbox[26] as the activation status of microglia can be visualized by its progressive morphological transformation from a highly branched to an ameboid cell shape (Fig. 2c).

The toolbox provides us with 59 shape features that describe the morphology of microglia, including 17 shape features with high discriminatory power such as sphericity, circularity, betweenness, segments per branch, or number of nodes. To determine which parameters best explained the variance, we performed a principal component analysis (PCA) for the 17 shape features with high discriminatory power. We found that PC1 explained the highest variance (52.8%; Fig. 2d) and therefore examined parameters clustered with PC1. The mean values of PC1 scores for controls, 1 concussion and 3 concussions showed that the mice that received 3 concussions had significantly different morphology in PC1 (p = 0.0084 one-way ANOVA and Dunnetts'test; Fig. 2e). We then focused on the morphologic parameters related to PC1, such as betweenness (node centrality), sphericity (compactness of the cell in 3D), number of segments per branch or number of nodes[26] and found that although a single concussion did not induce major changes in microglia morphology, repetitive concussions did. Repetitive concussions induced long-lasting changes and activation of microglial cells, which become spherical and lose their branching to adopt an ameboid morphology (Kruskal–Wallis and Dunn's test; Fig. 2f–i). Interestingly, although these changes were significant, they were relatively subtle compared to the changes induced by more massive injuries such as moderate head trauma or stroke[26–28].

## Functional microglial changes indicate a specific turnover of excitatory synapses following repetitive concussions in the underlying cortex

Since microglial cells are significantly activated especially after repetitive concussions, we wanted to find out whether their function is also affected in the cortex underlying the concussion(s). To this end, we investigated whether and when microglia in the cortex can engulf synapses after single or repetitive concussions, as this is known to occur during the physiological refinement of circuits in the developing and mature CNS[29–33]. To do so, we perfused mice at different time points following single and repetitive concussions and immunostained brain sections for the microglial marker Iba1, the presynaptic excitatory marker vGlut, the presynaptic inhibitory marker vGat and the lysosomal marker CD68 and analyzed the pattern of engulfment (Fig. 3a). Our analysis of 3D reconstructions of confocal microscopy images with Imaris showed a transient significant increase in the volume of CD68-positive lysosomes in microglia 48 h and 1 week after injury, indicating an increased phagocytosis capacity of microglia after repetitive concussions (Kruskal–Wallis test with Dunn's multiple comparison tests; Fig. 3b–e). In comparison, the volume of CD68-positive lysosomes was increased only at 1 week following a single concussion. We also analyzed the number of vGlut-positive excitatory presynaptic inputs localized inside cortical microglia. Interestingly, we showed that although an increased amount of vGlut-positive synapses within microglia was found only at 1 week following a single concussion, it was observed earlier following repetitive concussions and was significantly prolonged in the cortex up to 6 weeks i.e. for the entire study period (Kruskal–Wallis test with Dunn's multiple comparison test) following injury (Fig. 3b–d, f). Remarkably, the amount of inhibitory marker vGat inside microglia was not significantly changed at any time after single or repetitive concussions (Fig. 3g–i). To determine whether the increased presence of excitatory presynaptic material inside microglia triggers a loss of excitatory synapses or is a demonstration of turnover of excitatory synapses, we examined the density of excitatory (vGlut) and inhibitory (vGat) synapses in the cortex 1 and 6 weeks after single and repetitive concussion

(Fig. 3j, k). We observed that the density of vGlut puncta is stable after a single concussion, whereas we found a marked – although not significant – decrease in synapse density after repetitive concussions at 1 week. After 6 weeks, vGlut synapse density was similar to control values for single and repetitive concussion. When we focused on vGat density, we could not detect a difference from control values at any of the time points considered (Fig. 3j, l). Taken together, these data suggest that a specific turnover of excitatory synapses occurs after repetitive concussions. Since we were able to detect changes using presynaptic markers, we also asked whether the post-synapse is altered. To this end, we used GFP-M mice and examined the spine density and morphology of pyramidal neurons in the cortex. We found no changes in spine density in either apical or proximal dendrites at any of the time points considered, e.g. 1 week, 3 weeks, or 6 weeks after the injuries (Supplementary Fig. 3a–f). To further characterize the changes in dendritic spines induced by single and repetitive concussions below the level of injury, we then characterized spine morphology as an indicator of the degree of spine maturation (Supplementary Fig. 4a–f). We divided spines into three types: mushroom, thin, and stubby spines with mushroom spines generally considered as the most stable and mature structures[34–36]. We investigated spine morphology in the apical tuft and proximal dendrite of control and single or repetitively injured pyramidal neurons in GFP-M mice following single and repetitive concussions (Supplementary Fig. 4b, c). We found that the proportion of the more stable mushroom spines and thin spines did not change at 1 week following either the single or the repetitive concussion.

In contrast, the proportion of the more immature stubby spines transiently increased at 1 week in the apical tuft (p = 0.0286; One-Way Anova and Dunnett's test) and proximal dendritic part (p = 0.0317; One-Way Anova and Dunnett's test) following repetitive but not single concussion (Supplementary Fig. 4d) arguing for a shortening of the neck and a destabilization of the spines. We then investigated spine morphology in the proximal dendrites and apical tuft at 3 and 6 weeks following the injuries and could not find any lasting changes (Supplementary Fig. 4d, e).

## Functional microglial changes are not limited to the cortex underlying the impact but are also found in the hippocampus

Since most of the observed behavioral impairment concerns learning and memory after repetitive concussions (Fig. 1), we thought to extend the analysis of functional microglial changes to the hippocampus (Fig. 4a). Here, we focused on 2 time points: 1 and 6 weeks to understand the early and sustained responses to single and repetitive concussions, and sampled microglia from 3 hippocampal regions - CA2, CA3 and dentate gyrus. When we examined the number of presynaptic excitatory inputs localized within hippocampal microglia, there was a significant increase after 1 week for the single concussion and after 6 weeks for the repetitive concussions, suggesting that the removal of excitatory synapses after repetitive concussions occurs on a longer time scale (Kruskal–Wallis test with Dunn's multiple comparison test; Fig. 4b–e). Interestingly, and analogous to the cortex, the inhibitory marker vGat in the hippocampus was not found more frequently inside the microglia after single or repetitive concussions at any time point (Fig. 4f). As in the cortex, we also investigated in the hippocampus, whether the increased engulfment of excitatory presynaptic material triggers a loss or indicates a turnover of excitatory synapses. To this end, we examined the density of excitatory (vGlut) and inhibitory (vGat) synapses sampled from the CA2, CA3, and the dentate gyrus regions of the hippocampus 1 and 6 weeks after single and repetitive concussions (Fig. 4g–i). Similar to the cortex, we observed a decreased density, albeit not significant, of excitatory input after 1 week, which recovered over time. No change was seen in inhibitory input density during the study period (Fig. 4g–i). These data support the turnover of excitatory input in the hippocampus, a region further away from the concussions that is important for memory and learning, after repetitive concussions.

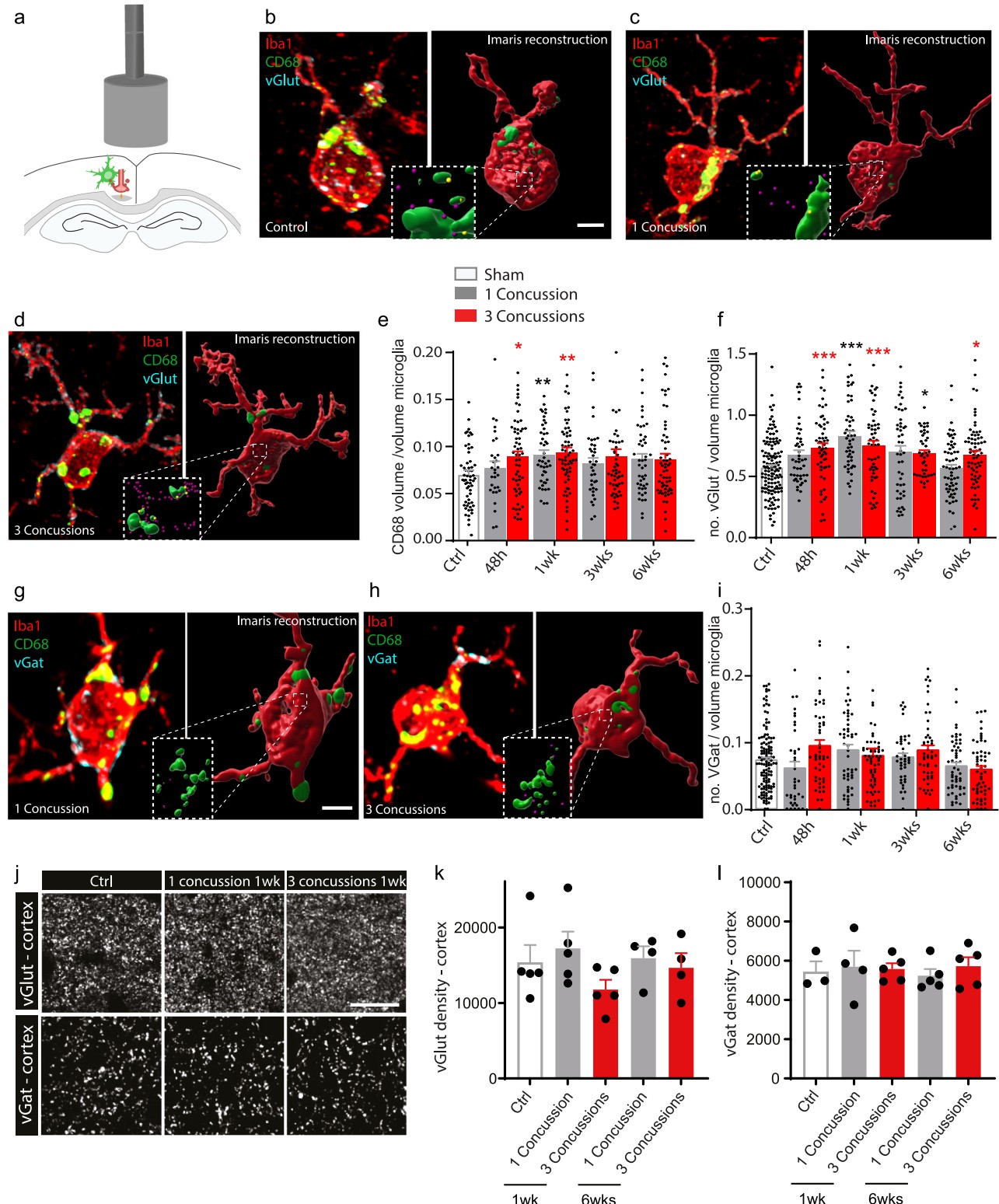

## Discussion

Here we demonstrate that microglial cells specifically engulf presynaptic excitatory inputs for several weeks after injury without decreasing the density of excitatory synapses. This suggests a specific increase in excitatory synapse turnover after repetitive concussions that is not seen after a single concussion and does not occur for inhibitory inputs. We also show that these changes occur not only in the cortical region underlying the impact but also in regions of the hippocampus. Since repetitive concussions also trigger long-lasting changes in learning ability, we suggest that the specific turnover of excitatory synapses can be compensated at an anatomical level but not at the behavioral level, likely because the new synapses are immature and weak.

### Repetitive concussions in mice cause cognitive and learning deficits rather than motor dysfunction

Most clinical concussions occur in the context of sports activities where the risk of repetition is increased. In this study, based on previous reports, we

**Fig. 3 | Increased amounts of presynaptic excitatory material are found in cortical microglia following repetitive concussions. a** Scheme of the experimental design investigating cortical microglia-mediated synapse engulfment in the cortex (green: microglia; red: presynaptic terminal). **b–d** Confocal images and 3D surface rendering in Imaris of excitatory synaptic marker (cyan, purple in 3D reconstructions) engulfment in lysosomes (green) of microglia (red) in controls (**b**), in single concussion at 6 weeks (**c**), in repetitive concussions at 6 weeks (**d**). **e** Quantification of volume of CD68–positive lysosome per volume microglia in the cortex of control mice and at different time points following injury. n = 4 (48 h) to 5 (1 wk, 3 wks, and 6 wks) mice per group, 30–62 microglia per group. P = 0.0185 Ctrl vs 3 concussions 48 h; p = 0.0053 Ctrl vs 1 concussion 1 wk; p = 0.0011 Ctrl vs 3 concussions 1 wk. **f** Quantification of the engulfment of vGlut excitatory synapses in the cortex of control mice and at different time points following injury. n = 4 (48 h) to 5 (1 wk, 3 wks, and 6 wks) mice per group, 42–130 microglia per group. P = 0.0006 Ctrl vs 3 concussions 48 h; p > 0.0001 Ctrl vs 1 concussion 1 wk; p = 0.0005 Ctrl vs 3 concussions 1 wk; p = 0.0466 Ctrl vs 3 concussions 3 wks; p = 0.0409 Ctrl vs 3 concussions 6 weeks. **g, h** Confocal images and 3D surface rendering in Imaris of

inhibitory synaptic marker (cyan, purple in 3D reconstructions) engulfment in lysosomes (green) of microglia (red) in single concussion at 6 weeks (**g**), in repetitive concussions at 6 weeks (**h**). **i** Quantification of the engulfment vGat inhibitory synapses in the cortex of control mice and at different time points following injury. n = 4 (48 h) to 5 (1 wk, 3 wks, and 6 wks) mice per group, 37–128 microglia per group. **j** Confocal images of vGlut (top) and vGat (bottom) puncta in layer II/III of the cortex underlying the concussion(s) in controls and at 1 week following single repetitive concussions. **k** Quantification of vGlut puncta in layer II/III of the cortex at 1 and 6 weeks following single or repetitive concussions. n = 4–5 animals per group. **l** Quantification of vGat puncta in layer II/III of the cortex at 1 and 6 weeks following single or repetitive concussions. n = 3–5 animals per group. All data were tested with the Shapiro-Wilk test. Data in (**e, f, i**) are not normally distributed and were then compared using a Kruskall–Wallis test followed by Dunn's test. Data in (**k, l**) are normally distributed and were analyzed using one-way ANOVA followed by Dunnett's multiple comparison test. Scale bars in (**b–d, g, h**) equal 5 μm and 30 μm in (**j**). Ctrl: Control; wk: week. Each dot represent one microglia in (**e, f, i**). Data as mean ± SEM.

---

used a mouse model of repetitive concussion in which a cushioned midline blow was delivered to the midline of the closed skull[18]. We calibrated our injuries to produce no macroscopic brain changes; specifically, we did not detect hemorrhage, edema, contusion, bruising, skull fracture, or cell death[6,18,37,38] and mirror repetitive human concussions, in which loss of consciousness may or may not occur after impact and imaging shows little or no change[39,40]. We demonstrated that apnea and righting reflex decreased with successive impacts, which is consistent with most previous reports of midline concussion[18,41,42]. It is worth noting that some reports have sometimes observed increased or prolonged apnea duration or righting reflex responses after repetitive concussions[41,43], which are likely specific to certain injury models, locations, and animal species.

After a single concussion, we did not observe any motor or cognitive changes such as learning or memory impairment. This is consistent with the clinical literature, which reports few to no sequelae of a single concussion in young adults[44,45] and in experimental models of mild brain injury[37]. However, it is known that young adults who sustain a single concussion are at higher risk for repetitive concussions[46,47] and therefore investigation of the consequences of repetitive concussions is necessary. After repetitive concussions, the mice in our model show no motor deficits, whereas they show marked impairments in learning and memory. This is again consistent with experimental and clinical data, which primarily show that repetition of concussions leads to a worse outcome than a single concussion[48] and that most impairments affect cognitive and learning abilities[49]. In our study, we show that mice that have suffered repetitive concussions develop transient changes in spatial memory and longer-lasting changes in their learning behavior. These features are commonly observed in patients who suffer repetitive blows to the head, for example during sports, and who may develop amnesia or neuropsychological changes in the period following the repetitive concussions[4,40,50–52]. Our results recapitulate the clinical situation and are consistent with other experimental reports in mice showing cognitive and anxiety-related behavioral changes[53]. Our paradigm does not allow us to determine whether cumulative injury or repetition of mild injury is the key to permanent pathology. As previous studies have indicated a window of vulnerability following an initial concussion[46,54], it is likely that concussion recurrence is the key factor in exacerbating functional outcomes in this study.

**Structural and functional microglial activation allows the specific engulfment of excitatory synapses after repetitive concussions**

In this study, we reveal the role of microglial cells as key players in excitatory synapses engulfment after repetitive concussions. We show that microglial cells are activated by repetitive concussions and that this activation is accompanied by specific engulfment of excitatory synapses. Contact and synapse refinement have been studied primarily during development, where glial cells have been implicated in the process of synapse removal and circuit shaping[30,33,47,55–58]. Here, we show that microglial structure and function are

specifically affected by repetitive concussions in a model of subthreshold injury that causes no visible damage. Such involvement of microglia and their ability to engulf synapses after brain injury has been reported previously, but in a model with moderate open skull injury[59]. Here, the authors showed that late after injury, there is ongoing synaptic degeneration driven by microglial phagocytosis of complement-opsonized synapses in both the ipsilateral and contralateral brain, and that complement inhibition interrupted the degenerative neuroinflammatory response and reversed cognitive decline, even when therapy was delayed for up to 2 months after traumatic brain injury. Our data extend these findings and identify excitatory synapses as the major vulnerable synapse type removed by microglial cells after repetitive concussions. Interestingly, both microglial activation and engulfment are only visible after repetitive concussions, suggesting that microglial activation and function are dependent on the repetition of concussions. These data suggest that the structural and specifically functional activation of microglia may be the key mediator causing the exacerbation of symptoms upon concussion recurrence, as previously reported[25]. Interestingly, the specific engulfment of excitatory synapses prompted us to investigate the density of excitatory and inhibitory synapses in the cortex and hippocampus. Here we observed a slight loss of synapses in the underlying cortex and hippocampus early after the repetitive concussions but not at chronic time points. This suggests a continuous turnover of excitatory synapses in the repetitively injured brain. While we did not survey the underlying white matter regions, it is known that repetitive concussions also trigger white matter abnormalities including microglia both experimentally and clinically[60,61]. Recently, acute changes in synaptic strength have been reported in the first few hours after mild brain injury[62]. Here, we report an early imbalance between excitatory and inhibitory input. This imbalance between excitatory and inhibitory input has been emphasized in other recent reports following concussion. In particular, persistent upregulation of excitatory synapses in the hippocampus, possibly through changes in postsynaptic Ca2+ signaling, was shown in hippocampal slices one month after single and repetitive concussions[63]. Langlois and colleagues have shown in a model with 3 repetitive closed concussions that repetitive concussions induce persistent changes in the spontaneous synaptic activity of hippocampal neurons[64]. Here, the authors suggest that repetitive concussions induce a shift in synaptic drive towards more inhibition in hippocampal CA1 neurons two weeks after repetitive concussions. Here, we extend these observations and show a specific turnover of presynaptic excitatory input that can be compensated for at the anatomical level by synaptic density, but not at the behavioral level, resulting in a persistent impairment of the animals' cognitive abilities. We hypothesize, here, that the turnover of excitatory synapses is towards immature and weak synapses[65] that need time to mature and be efficient, and which may be particularly vulnerable in subsequent concussion. Our study also suggests changes at the post-synaptic level. Using GFP-M mice with specific fluorescent labeling (GFP expression under the control of the thy1 promoter[66] of a subset of

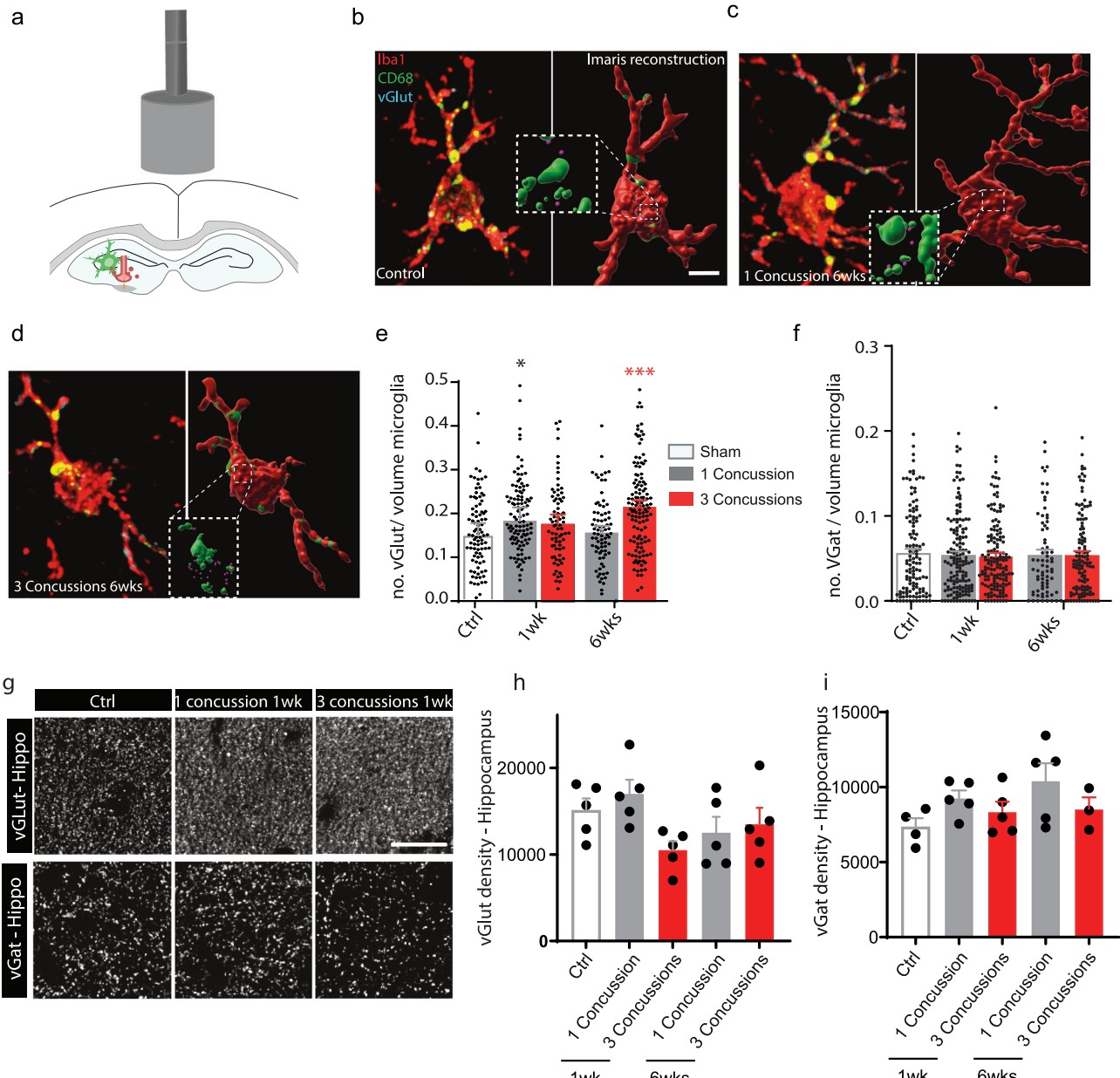

**Fig. 4 | Increased amounts of presynaptic excitatory input are also present in hippocampal microglia following repetitive concussions. a** Scheme of the experimental design investigating hippocampal microglia-mediated synapse engulfment in the hippocampus (green: microglia; red: presynaptic terminal). **b** Confocal images and 3D surface rendering in Imaris of synaptic marker (cyan, purple in 3D reconstructions) engulfment in lysosomes (green) of microglia (red) in controls. **c** Confocal images and 3D surface rendering in Imaris of synaptic marker (cyan, purple in 3D reconstructions) engulfment in lysosomes (green) of microglia (red) in single concussion at 6 weeks. **d** Confocal images and 3D surface rendering in Imaris of synaptic marker (cyan, purple in 3D reconstructions) engulfment in lysosomes (green) of microglia (red) in repetitive concussions at 6 weeks. **e** Quantification of the engulfment of vGlut excitatory synapses in the hippocampus of control mice and at different time points following injury. n = 5 (1 wk and 6 wks) mice per group, 78–127 microglia per group. p = 0.0135 Ctrl vs 1 concussion 1 wk

and p < 0.0001 Ctrl vs 3 concussions 6 wks. **f** Quantification of the engulfment of vGat inhibitory synapses in the hippocampus of control mice and at different time points following injury. n = 5 (1 wk and 6 wks) mice per group, 73–152 microglia per group. **g** Confocal images of vGlut (top) and vGat (bottom) puncta in the hippocampus in controls and at 1 week following single repetitive concussions. **h** Quantification of vGlut puncta in the hippocampus at 1 and 6 weeks following single or repetitive concussions. n = 5 animals per group. **i** Quantification of vGat puncta in the hippocampus at 1 and 6 weeks following single or repetitive concussions. n = 3–5 animals per group. All data were tested with the Shapiro–Wilk test. Data in (**e**, **f**) are not normally distributed and were then compared using a Kruskall–Wallis test followed by Dunn's test. Data in (**h**, **i**) are normally distributed and were analyzed using one-way ANOVA followed by Dunnett's multiple comparison test. Scale bars in (**b–d**) equal 5 μm and 30 μm in (**g**). Ctrl: Control; wk: week. Each dot represent one microglia (**e**, **f**). Data as mean ± SEM.

pyramidal neurons in the cortex and hippocampus), we showed that although there was no change in spine density, there were large and transient changes in spine morphology with more immature spines 1 week after repetitive concussions. Since the changes at the pre- and postsynaptic level can only be detected in repetitive concussions but not in single concussions, this emphasizes the vulnerability of the already injured brain to subsequent

impairments. One limitation of the current study is that we did not get to uncover the molecular pathway driving the excitatory synapse engulfment. While the complement pathway may mediate this engulfment - as previously demonstrated by Alawieh and colleagues for moderate brain injury[59] - this remains to be demonstrated following repetitive concussions. Understanding the molecular regulators of such microglia activation in the

context of repetitive concussions might allow early intervention within the window of brain vulnerability and prevent worsening outcomes following a second or third repetition of concussions.

## Materials and methods

### Animals

All the experiments were carried out under the approval of the Regierung von Oberbayern under the protocol number Vet_20-164. We have complied with all relevant ethical regulations for animal use. Eight to twelve weeks-old C57Bl6J (Janvier Labs) and GFP-M mouse line[66] mixed sex (predominantly female), were used for this study. All animals were housed under controlled standard housing conditions (dark/light cycle of 12 h, temperature 22 ± 2 degrees, and humidity of 55 ± 10%) with food and water ad libitum.

### Single and repetitive concussions using the controlled cortical impact (CCI)

Mice were anesthetized using 3% isoflurane (Piramal) delivered through a chamber (Harvard Apparatus, Holliston, USA) After being put in a stereotaxic frame (Precision Systems & Instrumentation, LLC), mice were maintained under 1,5% isofluorane during the whole procedure until the impact. A TBI-0310 Impactor (Precision Systems & Instrumentation, LLC) using a 5 mm diameter silicone tip was used to administer an impact of a 3.5 m/s speed, 2.0 mm depth and 500 ms dwell time[18]. The landing position of the tip spanned the entire skull with Lambda as a caudal landmark, aligned with the midline impacting both hemispheres equally and widely (Fig. 1). Following the impact, apnea and righting reflex duration (time needed for each mouse to get back to their prone position) were assessed on the individual mouse to assure reproducibility of the injury. Mice were put back on a heating pad until completion of the experiment. For pain management, meloxicam (Metacam, Boehringer Ingelheim) was administered. The repetitive concussion group underwent 3 impacts spaced 48 h apart, the single concussion group received only one impact and the Sham group was subjected to the same procedure, undergoing anesthesia and subsequently put into the stereotaxic frame without any impact. No mice were excluded from the study. The weight of the mice was recorded every day for the duration of the experiments.

### Behavioral tests and processing

Mice were habituated 3 times before any initial testing and were recorded for baseline before they underwent concussion. Cognitive and anxiety tests such as Y-maze, and tail suspension were carried out as well as motor tests such as the Rotarod and ladder rung. All the tests have been carried out by two observers blinded with respect to injury and time points.

**Y-maze test**. The Y maze was used to assess the spatial memory of each mouse and their spontaneous alternation potential. The spontaneous alternation was used to test the spatial memory. In-house custom-made Y-maze was used to perform the test. Y-maze comprises 3 separate arms (8 × 30 × 15 cm, at a 120° angle from one another). The test was performed in two steps: in the initial phase, the 'conditioning' phase, a random arm was closed and another was randomly picked as the 'starting' arm. The mouse was placed on the 'starting' arm and was left to explore the two arms for 5 min. In the test phase, after 30 min, the mouse was returned to the 'starting' arm and was allowed to explore the entire Y-maze for 5 min. In between each conditioning and test phase, the Y-maze was cleaned with 70% ethanol. Parameters were recorded with a GoPro 8 camera at 120 frames per second. The preference index was calculated as follows: (time in the novel arm− time in the old arm)/(time in the novel arm+ time in the old arm)[67].

**Tail suspension**. Tail suspension is used to assess anxiety, behavioral despair, and learned helplessness[22,23]. Mice were placed individually and isolated from each other in a self-made box and suspended by their tail at 30 cm above the bench. As mice have been shown to climb on their tail

during the procedure, a tube covering the mouse's tail was placed to avoid this behavior. The trial lasted 6 min and each trial was recorded with a GoPro 8 camera. The motility and immobility of each mouse were assessed using Behavioral Observation Research Interactive Software (BORIS)[68].

**Ladder rung**. For assessment of regular walking and fine paw placement, the ladder rung test was used[21,69]. In this test, the animals have to cross a 1 m horizontal grid ladder 3 times consecutively and footfalls are counted by an investigator blinded to injury and time point based on video recordings. We evaluated locomotion with evenly distributed spacing between the rungs and the animal's ability for fine-coordinated paw placements using irregular spacing of the rungs. We only analyzed consecutive steps and the last step before or after an interruption was therefore not evaluated. Footfalls were recorded when mice either totally missed a rung or if they slipped from a rung (deep or slight slip). The number of footfalls was calculated quantitatively for a given distance.

**Rotarod test**. To evaluate motor ability and balance, we used the rotarod test (Ugo Basile, Italy)[19]. Mice were placed on the apparatus which was kept either at a constant speed of 20 rounds/min or accelerated from 2 to 40 rounds/min. The maximum score is 120 and the device automatically records the time and velocity at which the mice fell from the rod.

### Tissue processing and Immunohistochemistry

Mice were sacrificed and transcardially perfused with 4% paraformaldehyde (PFA) in 1 M PBS. The tissue was postfixed overnight and then microdissected.

**Microglia density and morphology**. For the evaluation of microglia density and morphology, mice's brains were embedded in a 3% agarose solution in deionized water. Sections of 100 μm were cut on a Leica VT 1000s vibratome. Sections were then stained following a two-day protocol. Initially, the sections, after being washed 3 times with 1× PBS for 10 min each, were mounted on slides. They were then incubated in a blocking solution of 10% horse serum (HS) in 0,5% (v:v) Triton X-100 in 1× PBS (T-PBS) for 1 h at room temperature (RT). The coronal sections were then incubated overnight at room temperature in a solution of 5% HS with T-PBS containing primary antibody: anti-rabbit anti-ionized calcium-binding adapter molecule-1 [IBA-1] (1/500, Wako, N°019-19741). After overnight incubation of the primary antibody, sections were washed 3 times and thereafter incubated for 2 h with a solution of 5% HS with T-PBS containing Alexa-594 donkey anti-rabbit (1/500, Thermo Fisher, A21207). Sections were then washed and coverslipped with VectaShield (Vector Laboratories). Analysis was performed in the sensory cortex (S1), layer II/III underlying the impact(s).

**Spine density and morphology**. To measure spine density and morphology we relied on the intense and sparse labeling of GFP-M mice[66]; essentially layer V neurons and fewer layer II/III neurons. No signal amplification was required. 100 μm sections were made using the vibratome. Sections were then washed and coverslipped with VectaShield (Vector Laboratories). Analysis was performed in the sensory cortex (S1), layer II/III underlying the impact(s).

**Microglia-mediated synaptic engulfment and density of excitatory and inhibitory synapses**. To evaluate the engulfment of presynaptic material by microglia and the density of excitatory and inhibitory synapses, brains were immersed in a 30% sucrose solution as cryopreservation for 48 h. The samples were then positioned into a cryo-mold, coated with Optimal Cutting Temperature compound (Tissue-Tek O.C.T. medium), and 30 μm thick sections were cut using a cryostat (Leica CM 1850 model). The sections were washed 1× PBS, followed by their incubation in a blocking solution (10% HS with T-PBS). The sections were then incubated with a solution of 5% HS with T-PBS

containing for our first set, guinea pig anti-IBA1 (1/500, Synaptic Systems, N°234 004), rabbit anti-VGLUT1/2 (1/500, Synaptic Systems, N°135 503), rat anti-CD68 (1/500, Abcam, N°ab53444). The second set consisted of rabbit anti-IBA1 (1/500, Wako, N°019-19741), guinea pig anti-VGAT (1/500, Synaptic Systems, N°131 308), rat anti-CD68 (1/500, Abcam, N°ab53444). After being incubated overnight, the section was incubated for 2 h containing secondary antibodies: anti-guinea pig Alexa Fluor (AF) 633 (1/500, Sigma Aldrich, N°SAB4600129), anti-rat AF594 (1/500, Thermo Fisher, N° A21209), anti-rabbit AF488 (1/500, Thermo Fisher, N°A21206), anti-rabbit AF647 (1/500, Thermo Fisher, N°A32795), anti-guinea pig Cy3 (1/500, Jackson Immunoresearch, N°706-165-148), anti-rat AF488 (1/500, Abcam, N°ab150153). The sections were thereafter incubated with a DAPI (1:10000) 1× PBS solution for 10 min. Sections were then washed and coverslipped with VectaShield (Vector Laboratories). Analysis was performed in the sensory cortex (S1), layer II/III underlying the impact(s), and in the stratum radiatum of the CA2, and CA3 regions, and in the dentate gyrus of the hippocampus.

**Neuronal density and cell death.** To assess the neuronal cell death and quantify neuron density, 30 μm sections were cut using a cryostat. Immunohistochemistry was performed as stated in the paragraph above. For cell death, we used an activated Casp3 antibody (1/250, Abcam, ab2302) coupled with an anti-rabbit AF647 (1/500, ThermoFisher, A32795): To determine neuronal density we used Neurotrace (NT; Thermofisher N21479). Sections were then washed and coverslipped with VectaShield (Vector Laboratories). Analysis was performed in the sensory cortex (S1), layer II/III underlying the impact(s).

### Image acquisition and processing
**Spine density and morphology.** In order to quantify the spine density/morphology of individual dendrites, we acquired z-stacks using the FV1000 Olympus microscope (60× oil objective NA 1.35, zoom of 3, and resolution of 1024 × 1024). We imaged two areas for each dendrite: proximal dendrite (at a distance of ~70–150 μm from the soma), and apical tuft (at a distance up to 75 μm below the S1 cortical surface). The images underwent deconvolution using the Huygens Essential software (16.05 Scientific Volume Imaging, NL), and Z projections of the deconvolved stacks were obtained using Neuron Studio.

**Microglia density, morphology, synapse density, and engulfment.** Leica SP8X WLL upright confocal microscope from the Bioimaging core facility from the biomedical center, Martinsried- Munich was used. Excitation was obtained using a continuous 405 nm laser and a pulsed white light laser (470–670 nm). Each section was scanned in a specific cortical region defined as follows: 100 μm from the midline and 20 μm depth from the surface of the coronal section. Each acquired z-stack was 20 μm thick each using a step size of 0,4 μm (NA 1.3, 40× oil, zoom 2 resolution 2048 × 2048, speed 600). Three to four sections were scanned per animal.

### Data analysis
**Synapse density.** Synapse density was quantified using Imaris software 9.6.0 version (BitPlane) in a semi-automatic manner. The spot feature parameters were preset for each Vglut and Vgat spot, by defining the diameter of each spot. Thereafter, these settings were used in a batch processing pipeline to 3D ender and quantify the synaptic markers accordingly. The ratios of excitatory/inhibitory markers were made to determine the imbalance of synapses following concussion.

**Spine density and morphology analysis.** Each deconvoluted stack containing the dendritic segment was opened in Neuronstudio (version 0.9.92) and the dendritic trunk was semi-automatically traced resulting in a series of green vertices superimposed on volume-rendered data (Supplementary Fig. 3). Thereafter, dendritic spines were detected and

classified. This automatic quantification was manually controlled to remove falsely detected spines, add falsely undetected, or reclassify misclassified spines (mushroom, thin, stubby). The obtained numbers of spines were divided by the length of the dendritic segment determined by the program.

**Microglia morphology.** To analyze microglia morphology, we used a published automatized analysis[26]. This method is based on 4 successive steps that comprise (i) preprocessing of the images, (ii) segmentation of microglia into nucleus, soma, and branches, (iii) creation of skeletons, and (iv) obtention of morphological features for the microglia. The pipeline is fully implemented in a MATLAB (MathWorks)-based script[26]. Parameters such as the sphericity, betweenness, the number of segments per branch, and the number of nodes were extracted. The analysis has been carried out in the cortex at layer II-III.

**Microglia synapse engulfment.** Microglia engulfment was conducted using Imaris software 9.6.0 version (BitPlane). At least ten individual Iba + cells per animal were reconstructed automatically in a stack of 5 μm. On each scanned section, each microglia was 3D reconstructed using the rendered surface feature and used as a mask on vGut and vGat channels and the spot function was used to identify the positive signals (spheres). The average intensity of five ROIs selected in the background of a random subset of images was used to define the threshold in each channel of the spheres and used in all the images. Spheres that were not fully engulfed were manually removed. Spheres were assessed on the CD68 channel to evaluate colocalization into phagocytic lysosomes. The CD68 channel was further masked by the cell surface and the volume was automatically reconstructed to identify the CD68 volume per cell.

**Neurotrace positive Cell count and cell death quantification.** Neurotrace (NT) 435/455 positive cells were quantified using Fiji (ImageJ, v.1.52 h) on three individual sections per mouse. This was carried out manually by a blinded investigator. To check for the viability of that neuron, activated cleaved caspase-3 double-positive cells were quantified using Fiji (ImageJ, v.1.52 h) on three individual sections per mouse.

### Statistics and reproducibility
All results are given as mean ± standard error of the mean (SEM) unless otherwise stated. Prism software (GraphPad, version 10.2.2) was used for statistical analysis. All datasets were initially tested for normality using the Shapiro-Wilk test. Parametric or non-parametric tests were used according to the Shapiro-Wilk normality test for the analysis of all datasets. In all analyses $p < 0.05$ was considered statistically significant and all significance levels are indicated in the figure legends as follows: *$p < 0.05$; **$p < 0.01$; ***$p < 0.001$. All data to reproduce the results are publicly available. https://doi.org/10.6084/m9.figshare.28359797 [70].

### Reporting summary
Further information on research design is available in the Nature Portfolio Reporting Summary linked to this article.

## Data availability
All datasets used and/or analyzed in this present study are deposited at Figshare under the following https://doi.org/10.6084/m9.figshare.28359797 [70].

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

## Acknowledgements

The authors would like to thank Bernadette Fiedler for excellent technical assistance, XiaoQian Ye for help with Matlab as well as Dana Matzek and Bianca Stahr for animal husbandry. We also thank the Core Bioimaging Facility of the Biomedical Center for their support and confocal imaging systems. FMB is supported by grants from the Deutsche Forschungsgemeinschaft (DFG): TRR274 Project ID 408885537; Munich Center for Systems Neurology (SyNergy; EXC 2145 / ID 390857198) and FOR 5705 (BA 4140/2-1). VVS is supported by a post-doctoral fellowship from the Humboldt foundation. We thank Laura Empl for help with illustrations.

## Author contributions

Design of experiments: F.B., M.C., J.M. Surgical procedures: M.C., J.M. Data collection and analysis: M.C., J.M., C.D., L.J., S.P.D., S.R.B., V.V.S. Figures: F.B.; S.P.D. Manuscript writing: F.B., M.C. All authors validated the final version of the manuscript.

## Funding

## Competing interests

The authors declare no competing interests.
