## [Transparent Peer Review file · Communications Biology]

Repetitive concussions promote microglia-mediated engulfment of presynaptic excitatory input associated with cognitive dysfunction

Corresponding Author: Professor Florence Bareyre

Version 0:

Reviewer comments:

Reviewer #1

(Remarks to the Author)

In this paper, the authors use an experimental mouse model of repetitive concussions. The work has potential to add to the current literature, addressing a gap in understanding the role of microglia in this type of injury. The authors first show the effects of the induced concussions (1x or 3x) on selected behavioral/cognitive outcomes (Fig 1 and several Supplemental Figs), specifically that multiple concussions may lead to learning problems. They then show data that they conclude indicates microglial activation (Fig 2) and two figures (Figs 3 and 4) of analysis of microglia engulfing selected synaptic markers in two brain regions. In all, the work is highly descriptive and does not have a strong impact. The authors claim that microglia are activated in their concussion model, but such a finding is not all that surprising and further, the data presented for this claim are underwhelming. In addition, the analysis of images showing localization of vGlut or vGat inside microglia have limited impact because there are no experiments done to show effects on excitatory (or inhibitory) synapses in the absence or alteration of microglia. The authors do not attempt to link the behavioral/cognitive results with that of selected microglial features, nor do they perform experiments to probe microglial function. Extensive experimental work and analysis would be required to increase the impact of this paper.

More specific comments follow:

In the Introduction, the text on microglia is phrased as if all microglial activation/response is detrimental to injury outcome, while many reports show instead that microglia limit acute pathology, limit secondary cell death, and promote healing.

Regarding data in Supplemental Fig 1, there could be effects/cell death prior to one week timepoint selected for analysis. It is possible that by one week later, glia (especially microglia) probably already responded and cell death occurred, with these cells cleared?

Data in Figure 2: Iba1 staining is not very good quality. The image shown in 2c does not represent "ameboid" morphology, in fact if anything this microglia has more complex branching. It is not clear if morphological analysis in 2g, h is taken from perimeter or field area of microglial cells as the images of microglia presented are not consistent with the results shown. Further, the data for microglial activation in Fig 2 is not all that impactful as it is expected that microglia would respond to brain injury.

Fig3 and 4 there are more data points in the analysis/results for vGlut than for vGat. Is it possible that differences in statistical power due to different sample sizes underlie the findings of "significance" for vGlut analysis but not vGat analysis? There are clear trends in the vGat results that are perhaps even more obvious than those in the vGlut datasets, but the vGat data has fewer points.

Fig 3 and 4, what does each datapoint represent? A single microglia that was rendered?

Fig 1c, lines 134-135 it seems that the large scale of brain viewing is not detailed enough to understand the impacts of the head injuries in their model.

Lines 153-154 This statement is not justified with analysis only at 1 week post injury or later.

Line 180 "bruising" is used but previous statements claim no bruising, suggest alternate word choice

Line 183 mislabeled as Supp fig 5 (should be Supp fig 2)

Several typos using commas (,) for p values instead of (.) in the Results text.

Line 286-288 Statement needs citation(s)

Line 456 statement—this was not at all demonstrated in this work.

Discussion Lines 501-502, nowhere in this paper do the authors show that microglia drive this process.

Reviewer #2

(Remarks to the Author)

This study is overall a well written manuscript. The authors demonstrated that 1. repetitive concussions x 3 (induced at 48 hours apart) in mice led to significant structural activation of microglia to amoeboid morphology and an increase in microglia mediated engulfment of presynaptic excitatory synapses in layers II/III of the S1 cortex and the hippocampus (CA2, CA3 and the dentate gyrus) which were not observed in single concussive impact mice and 2. repetitive injured mice demonstrated spatial memory deficits on the Y-maze which was not observed in single impact mice. Several clarifications would help enhance the understanding of this study:

1. The authors used various ages of mice- 8 to 12 weeks of age; the authors should comment on whether the various ages may affect their study
2. Were there any effects on gender? The authors should provide this data.
3. Because this study evaluated very focused regions of the cortex for the morphologic analysis- S1, layers II/III and hippocampus (CA2, CA3, dentate gyrus), a more precise and detailed description of where they examined the microglial changes would be appreciated; for example, in the CA3 region, did the authors evaluate the stratum oriens vs. pyramidale vs. radiatum regions? Also, concussion studies commonly affect the underlying white matter regions, the authors should mention this in their discussion section.
4. For cell death (caspase) studies, why did the authors only evaluate the cortex and not the hippocampus?
5. In the Results sections, please carefully check and correct multiple typos related to p values- many have commas instead of periods and please also carefully check the Figure legends for typing and spacing errors

Version 1:

Reviewer comments:

Reviewer #1

(Remarks to the Author)

The authors have addressed the majority of comments from the first round of review. Limitations still remain in the impact of the study because no microglial manipulations were performed or demonstrated to have effects on the measured outcomes (synaptic densities, behavior of animals). There are some areas where some further wording changes are suggested:

Fig 3 and text/data:

There is claimed engulfment of synapses but the authors cannot be certain this is the origin of synaptic markers inside microglia, unless experimentally demonstrated or directly observed. Need to temper statements to reflect this uncertainty, for example using wording such as "localized to microglial bodies", "inside microglia", "associated with the lysosomal marker", etc. Some of these statements pertain to:

Line 311, 312 Fig 3 title, Fig 4 title

Reviewer #2

(Remarks to the Author)

I am satisfied with the revisions

Point by point response

COMMSBIO-24-3796 "Repetitive concussions promote microglia-mediated engulfment of presynaptic excitatory input and cognitive dysfunction".

Reviewer 1:

In this paper, the authors use an experimental mouse model of repetitive concussions. The work has potential to add to the current literature, addressing a gap in understanding the role of microglia in this type of injury. The authors first show the effects of the induced concussions (1x or 3x) on selected behavioral/cognitive outcomes (Fig 1 and several Supplemental Figs), specifically that multiple concussions may lead to learning problems. They then show data that they conclude indicates microglial activation (Fig 2) and two figures (Figs 3 and 4) of analysis of microglia engulfing selected synaptic markers in two brain regions. In all, the work is highly descriptive and does not have a strong impact. The authors claim that microglia are activated in their concussion model, but such a finding is not all that surprising and further, the data presented for this claim are underwhelming. In addition, the analysis of images showing localization of vGlut or vGat inside microglia have limited impact because there are no experiments done to show effects on excitatory (or inhibitory) synapses in the absence or alteration of microglia. The authors do not attempt to link the behavioral/cognitive results with that of selected microglial features, nor do they perform experiments to probe microglial function. Extensive experimental work and analysis would be required to increase the impact of this paper.

We thank the reviewer for the review process and have now added substantial new data and analysis to support our paper. All changes are tracked in yellow in the revised manuscript.

1. In the Introduction, the text on microglia is phrased as if all microglial activation/response is detrimental to injury outcome, while many reports show instead that microglia limit acute pathology, limit secondary cell death, and promote healing.

This is indeed correct. We have therefore changed our statement to reflect better the dual role of microglia in pathological context. We have included these in our introduction in the revised manuscript (**lines 71-79**).

2. Regarding data in Supplemental Fig 1, there could be effects/cell death prior to one week timepoint selected for analysis. It is possible that by one week later, glia (especially microglia) probably already responded and cell death occurred, with these cells cleared?

This is of course a possibility. We have therefore performed a new experiment aiming to analyze cell death at 2 days following single and repetitive concussion. We show that also at earlier time points following single and repetitive concussions we could not detect significant cell death. We have now included these new data to the **supplementary figure 1 and lines 151-156**.

3. Data in Figure 2: Iba1 staining is not very good quality. The image shown in 2c does not represent "ameboid" morphology, in fact if anything this microglia has more complex branching. It is not clear if morphological analysis in 2g, h is taken from perimeter or field area of microglial cells as the images of microglia presented are not consistent with the results

shown. Further, the data for microglial activation in Fig 2 is not all that impactful as it is expected that microglia would respond to brain injury.

We have now replaced the images of the microglia in 2a with better confocal images of microglia. We apologize for the picture in 2c that was exchanged between the control and 3 concussions. Indeed, the ramified microglia are the homeostatic ones that can be found in control animals. We have now rectified this. While we agree that microgliosis and changes in structures of microglia is not surprising following injury, we believe that differential activation following single or repetitive concussion is intriguing as it demonstrates a priming of these microglia following mild repetitive insults lasting for a prolonged time following the injury. This demonstrates that there can be a window of therapeutic intervention to revert their activation. To perform the morphology quantifications we used a published toolbox (Heindl et al., 2018). This automated quantification is based on the segmentation of microglial cells into nucleus, soma, and branches and on the construction of a skeleton that represents the spatial structure of cell bodies and branches. Then the automated software extracts the morphological features using properties derived from the surface area, volume, and skeleton of the cells. Importantly, this software has been validated using a manual quantification of microglia morphology using the Sholl analysis to generate ramification data. We have now added more information to the methodology to better describe how the analysis was performed **lines 712-718**. Interestingly we see that microglia following repetitive concussions behave very similarly to microglia in peri-infarct areas following stroke although the activation seems much lower following concussions. This is not so surprising considering the mildness of the damage compared to a stroke model.

4. Fig3 and 4 there are more data points in the analysis/results for vGlut than for vGat. Is it possible that differences in statistical power due to different sample sizes underlie the findings of “significance” for vGlut analysis but not vGat analysis? There are clear trends in the vGat results that are perhaps even more obvious than those in the vGlut datasets, but the vGat data has fewer points.

Thanks for this comment. For the cortex, we have evaluated the engulfment of vGlut synapses in 566 microglia and vGat synapses in a total of 587 microglia in the cortex. These were spread over the different conditions and time points as depicted in reviewer figure 1 below.

Reviewer Figure 1. Number of microglia evaluated for engulfment analysis of vGlut or vGat synapses in the cortex. A total of 566 microglia was evaluated for the vGlut analysis while a total of 587 microglia was evaluated for the vGat analysis. Ctrl: Control; Cc: Concussion; hrs: hours; wk: week; wks: weeks.

For the hippocampus, we have evaluated 485 microglia for vGlut engulfment and 431 microglia for vGat engulfment (fewer time points compared to the cortical data). We have also compared the number by categories as presented in Reviewer Figure 2 below.

Reviewer Figure 2. Number of microglia evaluated for engulfment analysis of vGlut or vGat synapses in the hippocampus. A total of 485 microglia was evaluated for the vGlut analysis while a total of 431 microglia was evaluated for the vGat analysis. Ctrl: Control; Cc: Concussion; hrs: hours; wk: week; wks: weeks

As observed the numbers are quite comparable between vGlut and vGat analysis within time points and single/repetitive concussions. However, in the cortex, for the 3 concussions at 1 week we evaluated indeed less microglia for vGat than for vGlut. In the hippocampus, for the 3 Concussions at 6 weeks and the 1 concussion 1 week, we also evaluated less microglia for vGat than for vGlut. We have therefore now performed new analysis of all these data sets making sure that we include enough microglia in the vGat analysis.

We have therefore now re-evaluated this data sets and evaluated more microglia for the vGat category (see Reviewer Table below). We find that with the completed analysis and the increased number of microglia evaluated, the data are not significantly changed and our conclusion can be maintained. We have now replaced the initial datasets for vGat with the new data sets that included more evaluated microglia in the **revised Figure 3 and Figure 4** and are now stating the number of microglia evaluated per group in the figure legend **lines 336-337, 343 and 447-448, 450.**

	Cortex		Hippocampus	
	vGlut engulfment	vGat engulfment	vGlut engulfment	vGat engulfment
Ctrl	130	152	91	122
1CC 48hrs	47	47	-	-
3CC 48hs	56	57	-	-
1CC 1week	50	58	108	73151
3CC 1week	57	3859	78	64152
1CC 3weeks	48	47	-	-
3CC 3weeks	42	42	-	-
1CC 6weeks	60	60	81	6977
3CC 6weeks	76	76	127	51121

Reviewer Table: Number of microglia evaluated before (black numbers) and after the revisions (red numbers) for engulfment of vGlut and vGat synapses in controls and following single concussion (1CC) or repetitive concussion (3CC).

5. Fig 3 and 4, what does each datapoint represent? A single microglia that was rendered?

Each data point represents one microglia. We have added this information in the figure legend **lines 351 and 4598**

6. Fig 1c, lines 134-135 it seems that the large scale of brain viewing is not detailed enough to understand the impacts of the head injuries in their model.

Thanks for this comment. We have now drawn red circles on top of the images to show the place and extend of the impact of the concussions on the brains. See **Figure 1c** and the figure legend **lines 115-116**.

7. Lines 153-154 This statement is not justified with analysis only at 1 week post injury or later.

We have now conducted this analysis also at 2 days following the last concussion and have toned down this comment **lines 154-156**

8. Line 180 “bruising” is used but previous statements claim no bruising, suggest alternate word choice

Thank you, the word should have been “concussion”. We have now replaced it **line182**.

9. Line 183 mislabeled as Supp fig 5 (should be Supp fig 2)

This has been corrected **line 187** Thanks.

10. Several typos using commas (,) for p values instead of (.) in the Results text.

This has now been corrected throughout the “results” section.

11. Line 286-288 Statement needs citation(s)

We have added citations.

12. Line 456 statement—this was not at all demonstrated in this work.

We have removed this statement.

13. Discussion Lines 501-502, nowhere in this paper do the authors show that microglia drive this process.

We have removed this statement and replaced it with a sentence directly matching our data **lines 505-506** of the revised manuscript.

Reviewer 2:

This study is overall a well written manuscript. The authors demonstrated that 1. repetitive concussions x 3 (induced at 48 hours apart) in mice led to significant structural activation of microglia to amoeboid morphology and an increase in microglia mediated engulfment of presynaptic excitatory synapses in layers II/III of the S1 cortex and the hippocampus (CA2, CA3 and the dentate gyrus) which were not observed in single concussive impact mice and 2. repetitive injured mice demonstrated spatial memory deficits on the Y-maze which was not observed in single impact mice.

We are happy to see that the reviewer believes that our work is important and interesting but also supports actual clinical observations with sound methods and data. All changes are tracked in yellow in the revised manuscript.

1. The authors used various ages of mice- 8 to 12 weeks of age; the authors should comment on whether the various ages may affect their study

In order to answer this query, we have selected key datasets from the paper and have separated the most important datasets according to age comparing mice 8-9 weeks-old with mice 10-12 weeks-old (**reviewer figure 3** below). Using this method, we could not see any differences in the outcome based on age for any datasets evaluated. We believe that in our study the various ages of mice used is not widely spread (only 4 weeks differences) and does not influence the biological outcomes examined as depicted in the various graphs below.

Reviewer Fig. 3. Effect of age on several key parameters such as time in conditioning arm (a), microglia density (b), microglia sphericity at 48 hours (c) and vGlut engulfment at 1 week (d). Data indicate that no significance differences could be found between mice ages 8-9 weeks and mice aged 10-12 weeks. Cc: concussion, wk: week, wks: weeks, h: hours; Ctrl: Control.

2. Were there any effects on gender? The authors should provide this data.

Thank you for this suggestion that is of course relevant in view of the literature on microglia differences both structurally and functionally based on sex (Bobotis et al., 2023; Han et al., 2021; Rahimian et al., 2019). All groups in our experiments are, however, predominantly composed of female mice (average of 80% of female animals for all our groups). Therefore, such comparisons cannot robustly be analyzed due to the low number of male mice. We have added this information to the material and method **line 571**.

Rahimian R, Cordeau P, Kriz J (2019) Brain Response to Injuries: When Microglia Go Sexist, *Neuroscience*, **405**,14-23.

Bobotis BC, Braniff O, Gargus M, Akinluyi ET, Awogbindin IO, Tremblay ME (2023) Sex differences of microglia in the healthy brain from embryonic development to adulthood and across lifestyle influences. *Brain Research Bulletin*,**202**,110752.

Han J, Fan Y, Zhou K, Blomgren K, Harris RA. (2021) Uncovering sex differences of rodent microglia. *J Neuroinflammation*. **18**(1):74.

3. Because this study evaluated very focused regions of the cortex for the morphologic analysis-S1, layers II/III and hippocampus (CA2, CA3, dentate gyrus), a more precise and detailed description of where they examined the microglial changes would be appreciated; for example, in the CA3 region, did the authors evaluate the stratum oriens vs. pyramidale vs. radiatum regions? Also, concussion studies commonly affect the underlying white matter regions, the authors should mention this in their discussion section.

We thank the reviewer for this comment and are now indicating in the methodology that the evaluation was done by equal sampling of microglia on the stratum radiatum part of the CA2 and CA3 regions as well as in the dentate gyrus region **lines 672- 673**. We also now discuss the fact the concussions commonly affect the underlying white matter regions **lines 531- 533** of the discussion.

4. For cell death (caspase) studies, why did the authors only evaluate the cortex and not the hippocampus?

We reasoned that the cortex was directly underlying the injury and if cell death would occur it would occur primarily in the cortex. Of course, the hippocampus might also be particularly susceptible to injury and therefore we also have now investigated cell death in the hippocampus at 2days following injury (earlier time point also requested by reviewer 1). At this time point, we did not find any cell death in the cortex or in the hippocampus. We have now added the new cortical data at 2 days post-injury to **supplementary figure 1** of the revised manuscript and **lines 156-158**.

5 In the Results sections, please carefully check and correct multiple typos related to p values-many have commas instead of periods and please also carefully check the Figure legends for typing and spacing errors

Thanks. We have corrected the mistakes.

Point by point response

COMMSBIO-24-3796A "Repetitive concussions promote microglia-mediated engulfment of presynaptic excitatory input associated with cognitive dysfunction".

Reviewer 1:

There are some areas where some further wording changes are suggested:

Fig 3 and text/data: There is claimed engulfment of synapses but the authors cannot be certain this is the origin of synaptic markers inside microglia, unless experimentally demonstrated or directly observed. Need to temper statements to reflect this uncertainty, for example using wording such as "localized to microglial bodies", "inside microglia", "associated with the lysosomal marker", etc. Some of these statements pertain to:

Line 311, 312 Fig 3 title, Fig 4 title

We have changed the wording for the figure legend titles of Figure 3 and Figure 4. We have also changed the wording throughout the results for Figures 3 and 4. Specifically the changes can be found lines 224-226; 230; 232; 271; 273; 276; 832-833; 862-863.